# LEARNING TO SPLIT FOR AUTOMATIC BIAS DETECTION

## ABSTRACT

Classifiers are biased when trained on biased datasets. As a remedy, we propose Learning to Split (`ls`), an algorithm for automatic bias detection. Given a dataset with input-label pairs, `ls` learns to split this dataset so that predictors trained on the training split *cannot generalize* to the testing split. This performance gap suggests that the testing split is under-represented in the dataset, which is a signal of potential bias. Identifying non-generalizable splits is challenging since we have no annotations about the bias. In this work, we show that the prediction correctness of each example in the testing split can be used as a source of weak supervision: generalization performance will drop if we move examples that are predicted correctly away from the testing split, leaving only those that are mispredicted. `ls` is task-agnostic and can be applied to any supervised learning problem, ranging from natural language understanding and image classification to molecular property prediction. Empirical results show that `ls` is able to generate astonishingly challenging splits that correlate with human-identified biases. Moreover, we demonstrate that combining robust learning algorithms (such as group DRO) with splits identified by `ls` enables automatic de-biasing. Compared to previous state-of-the-art, we substantially improve the worst-group performance (23.4% on average) when the source of biases is unknown during training and validation. Our code is included in the supplemental materials and will be publicly available.

## 1 INTRODUCTION

Recent work has shown promising results on de-biasing when the sources of bias (e.g., gender, race) are known a priori Ren et al. (2018); Sagawa et al. (2019); Clark et al. (2019); He et al. (2019); Mahabadi et al. (2020); Kaneko & Bollegala (2021). However, in the general case, identifying bias in an arbitrary dataset may be challenging even for domain experts: it requires expert knowledge of the task and details of the annotation protocols Zellers et al. (2019); Sakaguchi et al. (2020). In this work, we study *automatic bias detection*: given a dataset with only input-label pairs, our goal is to detect biases that may hinder predictors' generalization performance.

We propose Learning to Split (`ls`), an algorithm that simulates generalization failure directly from the set of input-label pairs. Specifically, `ls` learns to split the dataset so that predictors trained on the training split *cannot generalize* to the testing split (Figure 1). This performance gap indicates that the testing split is under-represented among the set of annotations, which is a signal of potential bias.

The challenge in this seemingly simple formulation lies in the existence of many trivial splits. For example, poor testing performance can result from a training split that is much smaller than the testing split (Figure 2a). Classifiers will also fail if the training split contains all positive examples, leaving the testing split with only negative examples (Figure 2b). The poor generalization of these trivial solutions arise from the lack of training data and label imbalance, and they do not reveal the hidden biases. To ensure that the learned splits are meaningful, we impose two regularity constraints on the splits. First, the size of the training split must be comparable to the size of the testing split. Second, the marginal distribution of the labels should be the similar across the splits.

Our algorithm `ls` consists of two components, *Splitter* and *Predictor*. At each iteration, the Splitter first assigns each input-label pair to either the training split or the testing split. The Predictor then takes the training split and learns how to predict the label from the input. Its prediction performance on the testing split is used to guide the Splitter towards a more challenging split (under the regularity

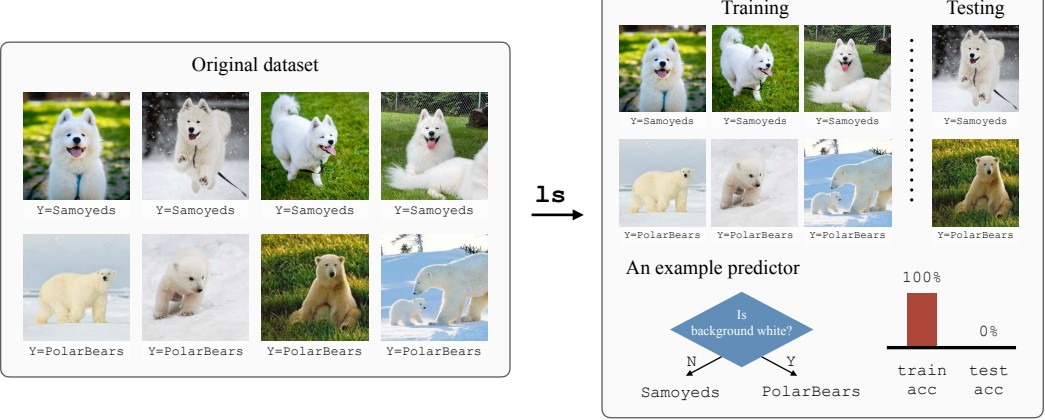

Figure 1: Consider the task of classifying samoyed images vs. polar bear images. Given the set of image-label pairs, our algorithm `ls` *learns* to *split* the data so that predictors trained on the training split *cannot generalize* to the testing split. The learned splits help us identify the hidden biases. For example, while predictors can achieve perfect performance on the training split by using the spurious heuristic: polar bears live in snowy habitats, they fail to generalize to the under-represented group (polar bears that appear on grass).

constraints) for the next iteration. Specifically, while we do not have any explicit annotations for creating non-generalizable splits, we show that the prediction correctness of each testing example can serve as a source of weak supervision: generalization performance will decrease if we move examples that are predicted correctly away from the testing split, leaving only those predicted incorrectly.

`ls` is task-agnostic and can be applied to any supervised learning problem, ranging from natural language understanding (Beer Reviews, MNLI) and image classification (Waterbirds, CelebA) to molecular property prediction (Tox21). Given the set of input-label pairs, `ls` consistently identifies splits across which predictors cannot generalize. For example in MNLI, the generalization performance drops from 79.4% (split by random) to 27.8% (split by `ls`) for a standard BERT-based predictor. Further analysis reveals that our learned splits coincide with human-identified biases. Finally, we demonstrate that combining group distributionally robust optimization (DRO) with splits identified by `ls` enables automatic de-biasing. Compared with previous state-of-the-art, we substantially improves the worst-group performance (23.4% on average) when the sources of bias are completely unknown during training and validation.

## 2 RELATED WORK

**De-biasing algorithms**   Modern datasets are often coupled with unwanted biases Buolamwini & Gebru (2018); Schuster et al. (2019); McCoy et al. (2019); Yang et al. (2019). If the biases have already been identified, we can use this prior knowledge to regulate their negative impact Kusner et al. (2017); Hu et al. (2018); Oren et al. (2019); Belinkov et al. (2019); Stacey et al. (2020); Clark et al. (2019); He et al. (2019); Mahabadi et al. (2020); Sagawa et al. (2020); Singh et al. (2021). The challenge arises when the source of biases is unknown (Li & Xu, 2021). Recent work has shown that the mistakes of a standard ERM predictor on its *training data* are informative of the biases (Bao et al., 2021; Sanh et al., 2021; Nam et al., 2020; Utama et al., 2020; Liu et al., 2021a; Lahoti et al., 2020; Liu et al., 2021b; Bao et al., 2022). They deliver robustness by boosting from the mistakes. (Wang & Vasconcelos, 2018; Yoo & Kweon, 2019) also utilize prediction correctness for confidence estimation and active learning. (Creager et al., 2021; Sohoni et al., 2020; Ahmed et al., 2020; Matsuura & Harada, 2020) further analyze the predictor's hidden activations to identify under-represented groups. However, many other factors (such as the initialization, the representation power, the amount of annotations, etc) can contribute to the predictors' training mistakes. For example, predictors that lack representation power may simply under-fit the training data.

In this work, instead of looking at the training statistics of the predictor, we focus on its *generalization gap* from the training split to the testing split. This effectively balances those unwanted factors.

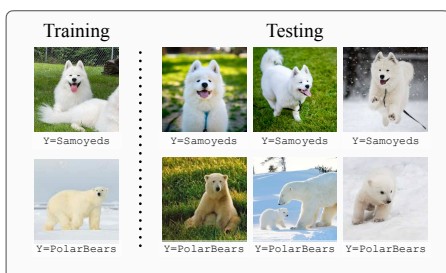 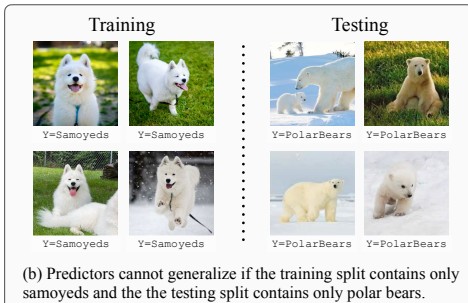

(a) Predictors cannot generalize if the size of the training split is incomparable to the size of the testing split.

(b) Predictors cannot generalize if the training split contains only samoyeds and the the testing split contains only polar bears.

Figure 2: Splits that are difficult to generalize do not necessarily reveal hidden biases. (a) Predictors cannot generalize if the amount of annotations is insufficient. (b) Predictors fail to generalize when the labels are unbalanced in training and testing. `ls` poses two regularity constraints to avoid such degenerative solutions: the training split and testing split should have comparable sizes; the marginal distribution of the label should be similar across the splits.

Going back to the previous example, if the training and test splits share the same distribution, the generalization gap will be small even if the predictors are underfitted. The gap will increase only when the training and testing splits have *different* prediction characteristics. Furthermore, instead of using a fixed predictor, we iteratively refine the predictor during training so that it faithfully measures the generalization gap given the current Splitter.

**Heuristics for data splitting**  Data splitting strategy directly impacts the difficulty of the under-lying generalization task. Therefore, in domains where out-of-distribution generalization is crucial for performance, various heuristics are used to find challenging splits Sheridan (2013); Yang et al. (2019); Bandi et al. (2018); Yala et al. (2021); Taylor et al. (2019); Koh et al. (2021). Examples include scaffold split in molecules and batch split for cells. Unlike these methods, which rely on human-specified heuristics, our algorithm `ls` learns how to split directly from the dataset alone and can therefore be applied to scenarios where human knowledge is unavailable or incomplete.

## 3  LEARNING TO SPLIT

### 3.1  MOTIVATION

Given a dataset $\mathcal{D}^{\text{total}}$ with input-label pairs $\{(x, y)\}$, our goal is to split this dataset into two subsets, $\mathcal{D}^{\text{train}}$ and $\mathcal{D}^{\text{test}}$, such that predictors learned on the training split $\mathcal{D}^{\text{train}}$ cannot generalize to the testing split $\mathcal{D}^{\text{test}}$.[1]

*Why do we have to discover such splits?*  Before deploying our trained models, it is crucial to understand the extent to which these models can even generalize within the given dataset. The standard cross-validation approach attempts to measure generalization by randomly splitting the dataset (Stone, 1974; Allen, 1974). However, this measure only reflects the *average* performance under the same data distribution $\mathbb{P}_{\mathcal{D}^{\text{total}}}(x, y)$. There is no guarantee of performance if our data distribution changes at test time (e.g. increasing the proportion of the minority group). For example, consider the task of classifying samoyeds vs. polar bears (Figure 1). Models can achieve good average performance by using spurious heuristics such as "polar bears live in snowy habitats" and "samoyeds play on grass". Finding splits across which the models cannot generalize helps us identify underrepresented groups (polar bears that appear on grass).

*How to discover such splits?*  Our algorithm `ls` has two components, a *Splitter* that decides how to split the dataset and a *Predictor* that estimates the generalization gap from the training split to the testing split. At each iteration, the splitter uses the feedback from the predictor to update its splitting decision. One can view this splitting decision as a latent variable that represents the prediction characteristic of each input. To avoid degenerate solutions, we require the Splitter to satisfy two regularity constraints: the size of the training split should be comparable to the size of the testing split (Figure 2a); and the marginal distribution of the label should be similar across the splits (Figure 2b).

---

[1]To prevent over-fitting, we held-out 1/3 of the training split for early-stopping when training the Predictor.

---

**Algorithm 1** ls: learning to split (see Algorithm 2 for full details)

---

**Input:** dataset $\mathcal{D}^{\text{total}}$
**Output:** data splits $\mathcal{D}^{\text{train}}, \mathcal{D}^{\text{test}}$

1: Initialize *Splitter* to random splitting
2: **repeat**
3:     Apply *Splitter* to split $\mathcal{D}^{\text{total}}$ into $\mathcal{D}^{\text{train}}, \mathcal{D}^{\text{test}}$. For each input-label pair $(x_i, y_i)$, sample the splitting decision $z_i \in \{0, 1\}$ from $\mathbb{P}_{Splitter}(z_i \mid x_i, y_i)$.
4:     Initialize *Predictor* and train *Predictor* on $\mathcal{D}^{\text{train}}$ using empirical risk minimization.
5:     Evaluate *Predictor* on $\mathcal{D}^{\text{train}}$ and $\mathcal{D}^{\text{test}}$. Compute generalization gap = difference in accuracy/AUC.
6:     **repeat**
7:         Sample a mini-batch from $\mathcal{D}^{\text{total}}$ to compute the regularity constraints $\Omega_1, \Omega_2$ (Eq 1).
8:         Sample another mini-batch from $\mathcal{D}^{\text{test}}$ to compute $\mathcal{L}^{\text{gap}}$ (Eq 2).
9:         Compute the overall loss $\mathcal{L}^{\text{total}} = \mathcal{L}^{\text{gap}} + \Omega_1 + \Omega_2$. Update Splitter to minimize $\mathcal{L}^{\text{total}}$.
10:    **until** $\mathcal{L}^{\text{total}}$ stops decreasing
11: **until** generalization gap stops increasing

---

### 3.2 SPLITTER AND PREDICTOR

Here we describe the two key components of our algorithm, *Splitter* and *Predictor*, in the context of classification tasks. The algorithm itself generalizes to regression problems as well.

**Splitter**   Given a list of input-label pairs $\mathcal{D}^{\text{total}} = [(x_1, y_2), \ldots, (x_n, y_n)]$, the Splitter decides how to partition this dataset into a training split $\mathcal{D}^{\text{train}}$ and a testing split $\mathcal{D}^{\text{test}}$. We can view its splitting decisions as a list of latent variables $\mathbf{z} = [z_1, \ldots, z_n]$ where each $z_i \in \{1, 0\}$ indicates whether example $(x_i, y_i)$ is included in the training split or not. In this work, we assume independent selections for simplicity. That is, the Splitter takes one input-label pair $(x_i, y_i)$ at a time and predicts the probability $\mathbb{P}_{Splitter}(z_i \mid x_i, y_i)$ of allocating this example to the training split. We can factor the joint probability of our splitting decisions as

$$\mathbb{P}(\mathbf{z} \mid \mathcal{D}^{\text{total}}) = \prod_{i=1}^{n} \mathbb{P}_{Splitter}(z_i \mid x_i, y_i).$$

We can sample from the Splitter's predictions $\mathbb{P}_{Splitter}(z_i \mid x_i, y_i)$ to obtain the splits $\mathcal{D}^{\text{train}}$ and $\mathcal{D}^{\text{test}}$. Note that while the splitting decisions are independent across different examples, the Splitter receives *global* feedback, dependent on the entire dataset $\mathcal{D}^{\text{total}}$, from the Predictor during training.

**Predictor**   The Predictor takes an input $x$ and predicts the probability of its label $\mathbb{P}_{Predictor}(y \mid x)$. The goal of this Predictor is to provide feedback for the Splitter so that it can generate more challenging splits at the next iteration.

Specifically, given the Splitter's current splitting decisions, we re-initialize the Predictor and train it to minimize the empirical risk on the training split $\mathcal{D}^{\text{train}}$. This re-initialization step is critical because it ensures that the predictor does not carry over past information (from previous splits) and faithfully represents the current generalization gap. On the other hand, we note that neural networks can easily remember the training split. To prevent over-fitting, we held-out 1/3 of the training split for early stopping. After training, we evaluate the generalization performance of the Predictor on the testing split $\mathcal{D}^{\text{test}}$.

### 3.3 REGULARITY CONSTRAINTS

Many factors can impact generalization, but not all of them are of interest. For example, the Predictor may naturally fail to generalize due to the lack of training data or due to label imbalance across the splits (Figure 2). To avoid these trivial solutions, we introduce two *soft* regularizers to shape the Splitter's decisions:

$$\begin{aligned}
\Omega_1 &= \mathrm{D}_{KL}(\mathbb{P}(z) \,\|\, \mathrm{Bernoulli}(\delta)), \\
\Omega_2 &= \mathrm{D}_{KL}(\mathbb{P}(y \mid z = 1) \,\|\, \mathbb{P}(y)) + \mathrm{D}_{KL}(\mathbb{P}(y \mid z = 0) \,\|\, \mathbb{P}(y)).
\end{aligned} \tag{1}$$

The first term $\Omega_1$ ensures that we have sufficient training examples in $\mathcal{D}^{\text{train}}$. Specifically, the marginal distribution $\mathbb{P}(z) = \frac{1}{n} \sum_{i=1}^{n} \mathbb{P}_{Splitter}(z_i = z \mid x_i, y_i)$ represents what percentages of $\mathcal{D}^{\text{total}}$ are split into $\mathcal{D}^{\text{train}}$ and $\mathcal{D}^{\text{test}}$. We penalize the Splitter if it moves too far away from the prior distribution $\text{Bernoulli}(\delta)$. Centola et al. (2018) suggest that minority groups typically make up 25 percent of the population. Therefore, we fix $\delta = 0.75$ in all experiments.[2]

The second term $\Omega_2$ aims to reduce label imbalance across the splits. It achieves this goal by pushing the label marginals in the training split $\mathbb{P}(y \mid z = 1)$ and the testing split $\mathbb{P}(y \mid z = 0)$ to be close to the original label marginal $\mathbb{P}(y)$ in $\mathcal{D}^{\text{total}}$. We can apply Bayes's rule to compute these conditional label marginals directly from the Splitter's decisions $\mathbb{P}_{S.}(z_i \mid x_i, y_i)$:

$$\mathbb{P}(y \mid z = 1) = \frac{\sum_i \mathbb{1}_y(y_i)\, \mathbb{P}_{S.}(z_i = 1 \mid x_i, y_i)}{\sum_i \mathbb{P}_{S.}(z_i = 1 \mid x_i, y_i)}, \quad \mathbb{P}(y \mid z = 0) = \frac{\sum_i \mathbb{1}_y(y_i)\, \mathbb{P}_{S.}(z_i = 0 \mid x_i, y_i)}{\sum_i \mathbb{P}_{S.}(z_i = 0 \mid x_i, y_i)}.$$

### 3.4 TRAINING STRATEGY

The only question that remains is how to learn the Splitter. Our goal is to produce *difficult* and *non-trivial* splits so that the Predictor cannot generalize. However, the challenge is that we don't have any explicit annotations for the splitting decisions.

There are a few options to address this challenge. From the meta learning perspective, we can back-propagate the Predictor's loss on the testing split directly to the Splitter. This process is expensive as it involves higher order gradients from the Predictor's training. While one can apply episodic-training (Vinyals et al., 2016) to reduce the computation cost, the Splitter's decision will be biased by the size of the learning episodes (since the Predictor only operates on the sampled episode). From the reinforcement learning viewpoint, we can cast our objectives, maximizing the generalization gap while maintaining the regularity constraints, into a reward function (Lei et al., 2016). However, according to our preliminary experiments, the learning signal from this scalar reward is too sparse for the Splitter to learn meaningful splits.

In this work, we take a simple yet effective approach to learn the Splitter. Our intuition is that *the Predictor's generalization performance will drop if we move examples that are predicted correctly away from the testing split, leaving only those that are mispredicted.* In other words, we can view the prediction correctness of the testing example as a direct supervision for the Splitter.

Formally, let $\hat{y}_i$ be the Predictor's prediction for input $x_i$: $\hat{y}_i = \arg\max_y \mathbb{P}_{Predictor}(y \mid x_i)$. We minimize the cross entropy loss between the Splitter's decision and the Predictor's prediction correctness over the testing split:

$$\mathcal{L}^{\text{gap}} = \frac{1}{|\mathcal{D}^{\text{test}}|} \sum_{(x_i, y_i) \in \mathcal{D}^{\text{test}}} \mathcal{L}^{\text{CE}}(\mathbb{P}_{Splitter}(z_i \mid x_i, y_i), \mathbb{1}_{y_i}(\hat{y}_i)). \tag{2}$$

Combining with the aforementioned regularity constraints, the overall objective for the Splitter is

$$\mathcal{L}^{\text{total}} = \mathcal{L}^{\text{gap}} + \Omega_1 + \Omega_2, \tag{3}$$

One can explore different weighting schemes for the three loss terms (Chen et al., 2018). In this paper, we found that the unweighted summation (Eq 3) works well out-of-the-box across all our experiments. Algorithm 1 presents the pseudo-code of our algorithm. At each outer-loop (line 2-11), we start by using the current Splitter to partition $\mathcal{D}^{\text{total}}$ into $\mathcal{D}^{\text{train}}$ and $\mathcal{D}^{\text{test}}$. We train the Predictor from scratch on $\mathcal{D}^{\text{train}}$ and evaluate its generalization performance on $\mathcal{D}^{\text{test}}$. For computation efficiency, we sample mini-batches in the inner-loop (line 6-10) and update the Splitter based on Eq equation 3.

---

[2]We note that the two regularizers $\Omega_1$ and $\Omega_2$ are introduced to shape the Splitter's decisions, but the model has the flexibility to deviate from this "prior." That is, the actual "posteriors" can be different depending on the dataset. For example, the minority group is unlikely to always constitute exactly 25% of the dataset. Therefore, it makes more sense to introduce soft regularizers instead of hard (and exact) constraints. Nevertheless, if users want to allocate exactly 25% of the data into the test set, instead of sampling from the Splitter's decisions $\mathbb{P}_{Splitter}(z_i \mid x_i, y_i)$, they can simply sort these probabilities and split at the 25th percentile.

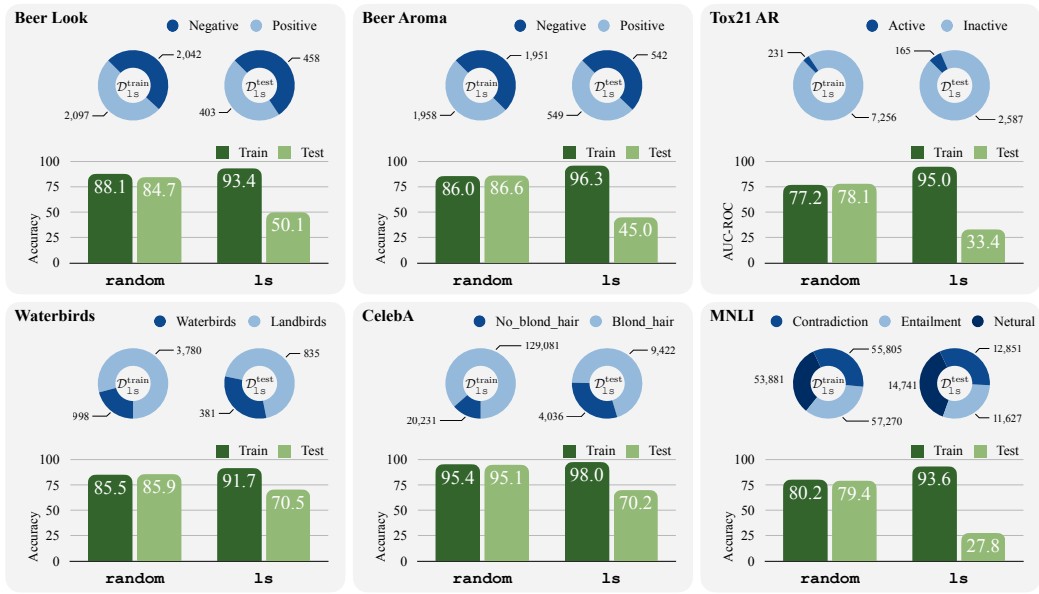

Figure 3: Even while the label distributions remain similar (blue), predictors that generalize on `random` splits fail to generalize on splits identified by `ls` (green). For both splits, to prevent over-fitting, we held-out 1/3 of the training split for early-stopping. In MNLI (lower right), the generalization gap for a standard BERT-based model is $93.6\% - 27.8\% = \mathbf{65.8\%}$.

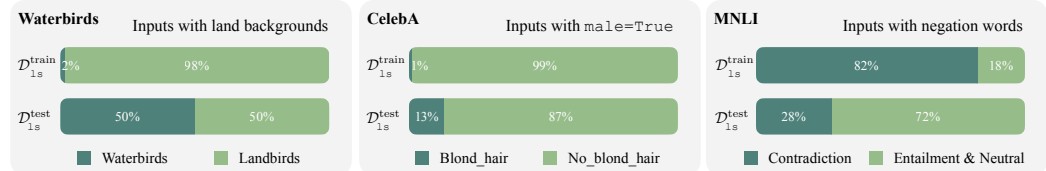

Figure 4: The splits learned by `ls` correlate with human-identified biases. For example in Waterbirds (left), `ls` learns to *amplify* the spurious association between landbirds and land backgrounds in the training split $\mathcal{D}^{\text{train}}$. As a result, predictors will over-fit the background features and fail to generalize at test time ($\mathcal{D}^{\text{test}}$) when the spurious correlation is reduced.

## 4 EXPERIMENTS

We conduct experiments over multiple modalities (Section 4.1) and answer two main questions. Can `ls` identify splits that are not generalizable (Section 4.2)? Can we use the splits identified by `ls` to reduce unknown biases (Section 4.3)? Implementation details are deferred to the Appendix. Our code is included in the supplemental materials and will be publicly available.

### 4.1 DATASET

**Beer Reviews** We use the BeerAdvocate review dataset (McAuley et al., 2012) where each input review describes multiple aspects of a beer and is written by a website user. Following previous work (Lei et al., 2016), we consider two aspect-level sentiment classification tasks: look and aroma. There are 2,500 positive reviews and 2,500 negative reviews for each task. The average word count per review is 128.5. We apply `ls` to identify spurious splits for each task.

**Tox21** Tox21 is a property prediction benchmark with 12,707 molecules Huang et al. (2016). Each input is annotated with a set of binary properties that represent the outcomes of different toxicological experiments. We consider the property Androgen Receptor (`active` or `inactive`) as our prediction target. We apply ls to identify spurious splits over the entire dataset.

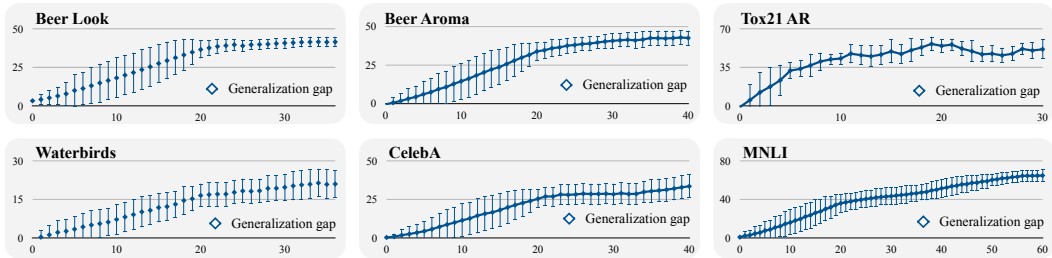

Figure 5: `ls`-identified splits correlate with certain spurious properties (`ATAD5`, `AhR`) even though they are not provided to algorithm. Here we present the train-test assignment of compounds with `AR=active` given by `ls`. In the leftmost bar, we look at all examples: 58% of {`AR=active`} is in the training split and 42% of {`AR=active`} is in the testing split. For each bar on the right, we look at the subset where an unknown property is active. For example, 17% of {`AR=active,ATAD5=active`} is allocated to the training split and 83% of {`AR=active,ATAD5=active`} is in the testing split.

Figure 6: Learning curve of `ls`. X-axis: number of outer-loop iterations. Y-axis: generalization gap from $\mathcal{D}^{\text{train}}$ to $\mathcal{D}^{\text{test}}$. Error bar represents the standard deviation across 5 random seeds.

**Waterbirds** Sagawa et al. (2019) created this dataset by combining bird images from the Caltech-UCSD Birds-200-2011 (CUB) dataset (Welinder et al., 2010) with backgrounds from the Places dataset (Zhou et al., 2014). The task is to predict waterbirds vs. landbirds. The challenge is that waterbirds, by construction, appear more frequently with a water background. As a result, predictors may utilize this spurious correlation to make their predictions. We combine the official training data and validation data (5994 examples in total) and apply `ls` to identify spurious splits.

**CelebA** CelebA is an image classification dataset where each input image (face) is paired with multiple human-annotated attributes Liu et al. (2015). Following previous work (Sagawa et al., 2019), we treat the hair color attribute ($y \in \{\text{blond}, \text{not\_blond}\}$) as our prediction target. The label is spuriously correlated with the gender attribute ({`male`, `female`}). We apply `ls` to identify spurious splits over the official training data (162,770 examples).

**MNLI** MNLI is a crowd-sourced collection of 433k sentence pairs annotated with textual entailment information (Williams et al., 2018). The task is to classify the relationship between a pair of sentences: entailment, neutral or contradiction. Previous work has found that contradiction examples often include negation words (McCoy et al., 2019). We apply `ls` to identify spurious splits over the training data (206,175 examples) created by Sagawa et al. (2019).

## 4.2 IDENTIFYING NON-GENERALIZABLE SPLITS

Figure 3 presents the splits identified by our algorithm `ls`. Compared to random splitting, `ls` achieves astonishingly higher generalization gaps across all 6 tasks. Moreover, we observe that the learned splits are not degenerative: the training split $\mathcal{D}^{\text{train}}$ and testing split $\mathcal{D}^{\text{test}}$ share similar label distributions. This confirms the effectiveness of our regularity objectives.

***Why are the learned splits so challenging for predictors to generalize across?*** While `ls` only has access to the set of input-label pairs, Figure 4 and Figure 5 show that the learned splits are informative of human-identified biases. For example, in the generated training split of MNLI, inputs with negation words are mostly labeled as contradiction. This encourages predictors to leverage the

Table 1: Average and worst-group test accuracy for de-biasing. When using bias annotations on the validation data for model selection, previous work (CVaR DRO (Levy et al., 2020), LfF (Nam et al., 2020), EIIL (Creager et al., 2021), JTT (Liu et al., 2021a)) significantly outperform ERM (that is also tuned using bias annotations on the validation data). However, they underperform the group DRO baseline (Sagawa et al., 2019) that was previously overlooked. When bias annotations are not available for validation, the performances of these methods quickly drop to that of ERM. In contrast, applying group DRO with splits identified by `ls` substantially improves the worst-group performance. † and ‡ denote numbers reported by Liu et al. (2021a) and Creager et al. (2021) respectively.

| Method | Bias annotated | | Waterbirds | | CelebA | | MNLI | |
| | in train? | in val? | Avg. | Worst | Avg. | Worst | Avg. | Worst |
| --- | --- | --- | --- | --- | --- | --- | --- | --- |
| Group DRO | ✓ | ✓ | 93.5%[†] | 91.4%[†] | 92.9%[†] | 88.9%[†] | 81.4%[†] | 77.7%[†] |
| ERM | ✗ | ✓ | 97.3%[†] | 72.6%[†] | 95.6%[†] | 47.2%[†] | 82.4%[†] | 67.9%[†] |
| CVaR DRO | ✗ | ✓ | 96.0%[†] | 75.9%[†] | 82.5%[†] | 64.4%[†] | 82.0%[†] | 68.0%[†] |
| LfF | ✗ | ✓ | 91.2%[†] | 78.0%[†] | 85.1%[†] | 77.2%[†] | 80.8%[†] | 70.2%[†] |
| EIIL | ✗ | ✓ | 96.9%[‡] | 78.7%[‡] | 89.5% | 77.8% | 79.4% | 70.0% |
| JTT | ✗ | ✓ | 93.3%[†] | 86.7%[†] | 88.0%[†] | 81.1%[†] | 78.6%[†] | 72.6%[†] |
| Group DRO (with supervised bias predictor) | ✗ | ✓ | 91.4% | **88.2%** | 91.4% | **88.9%** | 79.9% | **77.7%** |
| ERM | ✗ | ✗ | 90.7% | 64.8% | 95.8% | 41.1% | 81.9% | 60.4% |
| CVaR DRO | ✗ | ✗ | — | 62.0%[†] | — | 36.1%[†] | 81.8% | 61.8% |
| LfF | ✗ | ✗ | — | 44.1%[†] | — | 24.4%[†] | 81.1% | 62.2% |
| EIIL | ✗ | ✗ | 90.8% | 64.5% | 95.7% | 41.7% | 80.3% | 64.7% |
| JTT | ✗ | ✗ | — | 62.5%[†] | — | 40.6%[†] | 81.3% | 64.4% |
| Group DRO (with splits identified by `ls`) | ✗ | ✗ | 91.2% | **86.1%** | 87.2% | **83.3%** | 78.7% | **72.1%** |

presence of negation words to make their predictions. These biased predictors cannot generalize to the testing split, where inputs with negation words are mostly labeled as entailment or neutral.

**Convergence and time-efficiency**  `ls` requires learning a new Predictor for each outer-loop iteration. While this makes `ls` more time-consuming than training a regular ERM model, this procedure guarantees that the Predictor faithfully measures the generalization gap based on the current Splitter. Figure 6 shows the learning curve of `ls`. We observe that the generalization gap steadily increases as we refine the Splitter and the learning procedure usually converges within 50 outer-loop iterations.

### 4.3 AUTOMATIC DE-BIASING

Once `ls` has identified the spurious splits, we can apply robust learning algorithms to learn models that generalize across the splits. Here we consider group distributionally robust optimization (group DRO) and study three well-established benchmarks: Waterbirds, CelebA and MNLI.

**Group DRO**  Group DRO has shown strong performance when biases are annotated (Sagawa et al., 2019). For example in CelebA, gender (`male`, `female`) constitutes a bias for predicting blond hair. Group DRO uses the gender annotations to partition the training data into four groups: {`blond_hair`,`male`}, {`blond_hair`,`female`}, {`no_blond_hair`,`male`}, {`no_blond_hair`,`female`}. By minimizing the *worst-group* loss during training, it regularizes the impact of the unwanted gender bias. At test time, we report the average accuracy and worst-group accuracy over a held-out test set.

**Group DRO with supervised bias predictor**  Recent work consider a more challenging setting where bias annotations are not provided at train time. CVaR DRO (Levy et al., 2020) up-weights examples that have the highest training losses. LfF (Nam et al., 2020) and JTT (Liu et al., 2021a) train a separate de-biased predictor by learning from the mistakes of a biased predictor. EIIL (Creager et al., 2021) infers the environment information from an ERM predictor and uses group DRO to promote robustness across the latent environments. However, these methods still access bias annotations on the validation data for model selection. With thousands of validation examples (1199 for Waterbirds, 19867 for CelebA, 82462 for MNLI), a simple baseline was overlooked by the community: learning a bias predictor over the validation data (where bias annotations are available) and using the predicted bias attributes on the training data to define groups for group DRO.

**Group DRO with splits identified by `ls`**  We consider the general setting where biases are not known during both training and validation. To obtain a robust model, we take the splits identified by `ls` (Section 4.2) and use them to define groups for group DRO. For example, we have four groups in CelebA: $\{\texttt{blond\_hair}, z = 0\}$, $\{\texttt{blond\_hair}, z = 1\}$, $\{\texttt{no\_blond\_hair}, z = 0\}$, $\{\texttt{no\_blond\_hair}, z = 1\}$. For model selection, we apply the learned Splitter to split the validation data and measure the worst-group accuracy.

**Results**  Table 1 presents our results on de-biasing. We first see that when the bias annotations are available in the validation data, the missing baseline Group DRO (with supervised bias predictor) outperforms all previous de-biasing methods (4.8% on average). This result is not surprising given the fact that the bias attribute predictor, trained on the validation data, is able to achieve an accuracy of 94.8% in Waterbirds (predicting the spurious background), 97.7% in CelebA (predicting the spurious gender attribute) and 99.9% in MNLI (predicting the presence of negation words).

When bias annotations are not provided for validation, previous de-biasing methods (tuned based on the average validation performance) fail to improve over the ERM baseline, confirming the findings of Liu et al. (2021a). On the other hand, applying group DRO with splits identified by `ls` consistently achieves the best worst-group accuracy, outperforming previous methods by 23.4% on average. While we no longer have access to the bias annotations for model selection, the worst-group performance defined by `ls` can be used as a surrogate (see Appendix C for details).

## 5  DISCUSSION

Section 4 shows that `ls` identifies non-generalizable splits that correlate with human-identified biases. However, we must keep in mind that bias is a *human-defined* notion. Given the set of input-label pairs, `ls` provides a tool for understanding potential biases, not a fairness guarantee. If the support of the given dataset doesn't cover the minority groups, `ls` will fail. For example, consider a dataset with only samoyeds in grass and polar bears in snow (no samoyeds in snow or polar bears in grass). `ls` will not be able to detect the background bias in this case.

We also note that poor generalization can result from label noise. Since the Splitter makes its decision based on the *input-label* pair, `ls` can achieve high generalization gap by allocating all clean examples to the training split and all mislabeled examples to the testing split. Here we can think of `ls` as a label noise detector (see Appendix D for more analysis). Blindly maximizing the worst-split performance in this situation will enforce the model to memorize the noise.

Another limitation is running time. Compared to empirical risk minimization, `ls` needs to perform second-order reasoning, and this introduces extra time cost (see Appendix C for more discussion). Finally, in real-world applications, biases can also come from many independent sources (e.g., gender and race). Identifying multiple *diverse* splits will be an interesting future work.

## 6  CONCLUSION

We present Learning to Split (`ls`), an algorithm that learns to split the data so that predictors trained on the training split cannot generalize to the testing split. Our algorithm only requires access to the set of input-label pairs and is applicable to general datasets. Experiments across multiple modalities confirm that `ls` identifies challenging splits that correlate with human-identified biases. Compared to previous state-of-the-art, learning with `ls`-identified splits significantly improves robustness.

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
