# OpenReview forum: "Learning to Split for Automatic Bias Detection"
_ICLR.cc/2023/Conference — Submitted to ICLR 2023_

### Official Review · Reviewer_Jd9v · 2022-10-20

**Confidence:** 4
**Correctness:** 1
**Technical Novelty And Significance:** 2
**Empirical Novelty And Significance:** 2
**Recommendation:** 5

**Clarity, Quality, Novelty And Reproducibility:**

Clarity - Overall the writing is clear.  The notation, however, could use elaboration. Specifically the equations in 3.3 and 3.4 could use some elaboration about specifically is meant by their notation.

Quality - Beyond the strengths and weaknesses above, I have no further notes on the quality of the work.

Novelty - Much of the proposed methodology is rather straight-forward and does not introduce many novel concepts. The main source of novelty of the work is in the posing of the problem itself: The task of splitting a data set so that a model achieves high generalization gap between test and train sets.  While i have not encountered work performing this exact task, a lot of work has focused on sources of bias.  Also, there is another parallel line of work on test and evaluation of ML models that aims to uncover biases present in data collections.

Reproducability - Algorithm 1 lacks any level of specificity in terms of how the sampler is used to split data and what optimization procedures are used to update the sampler or predictor.  As such, I feel it is very difficult to reproduce this work.

**Strength And Weaknesses:**

Strengths
1. The problem of automatically identifying biases in data sets is clearly one of major practical importance to the use of machine learning techniques in real-world setting.
2. Their technique is conceptually simple, and seems rather easy to implement.
3. The results reported clearly show LS is able to split data to increase the generalization gap compared to random sampling on a large number of varied data sets.

Weaknesses
1. I think the problem as set up in the paper may lack some practical utility.  I do not see the direct connection bween intentionally biasing train and test splits and measuring the generalization gap and bias that can arise when training a model on the data set.  What is not clear is that if a normal training procedure which randomly splits train and test data results in a sufficiently accurate model on the subpopulation discovered by LS.  If so, then LS is measuring a worst case generalization performance that may not be ever realized.  I think a more compelling set up would be if a training data is representative of a standard split, and a subpopulation can be found within a test set for which a model performs poorly.  This is much more representative of how data is used to train a model.
2. I think the experiments could benefit from other baseline lines than random splits.  For instance, in the data sets where auxiliary attributes are available (like CelebA), you could split the train/test data across attributes and measure the generalization gap.  Doing this can result in different “oracle” baselines that better explain generalization gap relative to clear biases.  Further, I feel a conceptually much simpler and heuristic approach can be taken based on the same principle as (2).  At each outer iteration, simply choose 25% of the samples, accounting for label distributions, that the predictor performed the worst on.  This doesn’t require a complex splitter model, can exactly satisfy the desired constraints, and is in the same spirit as LS.
3. The regularization approach seems inexact for a sampling procedure, and the constraints attempted to be imposed by the regularization terms are not satisfied in the experiments (~82% of the examples are in the train set for the “Beer Look” experiment).  It’s not clear how important this is to be exact but if it is important, then stronger enforcement of constraints needs to be considered.
4. Overall the paper doesn’t provide much of principled framing of the problem being solved nor the proposed technique.  For instance, since label distributions are controlled for in LS, it would seem the cause of generalization gap is due to either concept drift (p(y|x) changing) or covariate shift (p(x) changing).  Without identifying the underlying formal definition of bias, it is difficult to understand what is being modeled in a principled manner.  Similarly, it is difficult to reason about how much LS is guided by distributional shifts in the data or the predictive power of the predictor (i.e. Is the generalization gap found by LS a function or the data, the model, the training procedure, or all of the above?).
5. The justification for the \delta parameter is taken out of context from Centola et al. 2018  Their work focuses on what proportion of a subpopulation is needed to tip a majority opinion to a minority opinion.  This has little to do with the problem that is the focus of this work.

Minor Points/Questions:
1. (2) can use some further elaboration.  As I understand it, the second term is and indicator if the predictor predicts the true label. This should be mentioned
2. There are no details on how (3) is optimized.  The free variables z_i are in {0,1}, which makes this not an direct application of gradient based methods as written.  How is it done?

**Summary Of The Paper:**

The authors focus on the problem of identifying potential generalization gaps that can be caused by selectively choosing train a test sets in supervised learning problems.  The authors propose an algorithm called Learning to Split (LS) that iteratively trains a model, uses a three part objective to train a "splitter" model to assign samples train and test sets, and then again retrains a model until the gap between train and test performance ceases to increase.  They show empirically that LS is able to find splits that have greater generalization gaps between train and test performance than randomly splitting the data.  Further, they show that methods that attempt to debias models work well when labels indicating subpopulations are given, but when not provided such labels, debiasing methods perform no better than standard empirical risk minimization.

**Summary Of The Review:**

Overall, I feel there are three major weaknesses of the paper that influenced my decision.
1.  I do not believe the specific problem formulation considered in this paper has much practical significance.  I am not convinced that intentionally splitting whole data sets to maximize a generalization gap necessarily means that training a model on the data set using standard data splits and procedures will be biased.
2. Due to lack of principled discussion and justification, I am not convinced that the generalization gap can entirely be explained by bias in the data and not biases from the specific predictor model or training procedure.
3. The experiments are quite limited in that there are baselines that should be used to justify the proposed approach.

(see rebuttal discussion below for more details on final scoring)

---

> ### Author Response · Authors · 2022-11-16
> **Thank you for your detailed comments (1/2)**
>
> *“I do not believe the specific problem formulation considered in this paper has much practical significance. I am not convinced that intentionally splitting whole data sets to maximize a generalization gap necessarily means that training a model on the data set using standard data splits and procedures will be biased.”*
>
> + It is common practice in many communities to define heuristic-based splits to properly measure generalization performance. So-called "standard splits" are often the result of such efforts.
>
> + This is necessary because it has been widely reported that directly training a model on randomly-split data will overfit to spurious biases (Table 1, and Levy et al., 2020; Nam et al., 2020; Creager et al., 2021; Liu et al., 2021a).
>
> + Given only input-label pairs, ls is an automatic, model-agnostic solution to identify such challenging splits for any supervised learning task.
>
> + These splits can then be used to enforce that the model optimizes for all groups, e.g. by minimizing worst-split loss. On three standard de-biasing benchmarks, our approach significantly improves the robust performance on the held-out data (Table 1).
>
>     | Worst-case accuracy    | Waterbirds | CelebA | MNLI  |
>     |------------------------|:----------:|:------:|:-----:|
>     | ERM                    | 64.8%      | 41.1%  | 60.4% |
>     | Group DRO on ls-splits | 86.1%      | 83.3%  | 72.1% |
>
> ---
>
> *"You could split the train/test data across attributes and measure the generalization gap. Doing this can result in different “oracle” baselines that better explain generalization gap relative to clear biases."*
>
> + The "average" and "worst-group" accuracies in Table 1 are the average and min over exactly these splits based on attributes.
>
> + In fact, we find strong correlation between ls-identified splits and human-defined oracle attributes in Figure 4 and 5.
>
> ---
>
> *"At each outer iteration, simply choose 25% of the samples, accounting for label distributions, that the predictor performed the worst on. This doesn’t require a complex splitter model, can exactly satisfy the desired constraints, and is in the same spirit as LS."*
>
> + A parametrized and learned splitter enables generalization across examples and flexible control over the resulting splits.
>   + In your proposed approach, since the predictor is trained on the training split, the examples that it performs the worst on will only come from the testing split. The proposed approach doesn't have the ability to move examples from the training split to the testing split.
>   + In Section 3.3, we propose two simple regularity constraints (split ratio constraint $\Omega_1$ and label marginal constarint $\Omega_2$) to avoid degenerative solutions. Users can also easily incorporate their customized constraints (e.g., biases that are orthogonal to gender) into this formulation. While your proposed heuristic approach may satisfy the split ratio constraint, it is unclear how to integrate these data-specific desiderata into the framework.
>
> ---
>
> *"The regularization approach seems inexact"*
>
> + The two regularizers $\Omega_1$ and $\Omega_2$ are introduced to shape the Splitter's decisions, but the model has the flexibility to deviate from this "prior." That is, the actual "posteriors" can be different depending on the dataset.
>
> + For example, the minority group is unlikely to always constitute exactly 25% of the dataset. Therefore, it makes more sense to introduce soft regularizers instead of hard (and exact) constraints.
>
> + Nevertheless, if users want to put exactly 25% of the data into the test set, instead of sampling from the Splitter's decisions $P(z_i|x_i,y_i)$, they can sort these probabilities and simply pick the top 25%.

---

> > ### Author Response · Authors · 2022-11-16
> > **Thank you for your detailed comments (2/2)**
> >
> > *"For instance, since label distributions are controlled for in LS, it would seem the cause of generalization gap is due to either concept drift (p(y|x) changing) or covariate shift (p(x) changing).  Without identifying the underlying formal definition of bias, it is difficult to understand what is being modeled in a principled manner."*
> >
> > + When the regularity constraints are satisfied, the generalization gap arises from the mismatch of $\mathbb{P}(x\mid y)$ between the learned splits. In other words, **both concept drift and covariate shift** can contribute to this gap.
> >
> > + For example in the Waterbird dataset (Figure 4), our learned training split consists mostly of landbirds with land backgrounds and waterbirds with water backgrounds. In the testing split, the numbers of waterbirds and landbirds are similar for both backgrounds. It is clear that both $\mathbb{P}(x)$ and $\mathbb{P}(y \mid x)$ are different across the two splits.
> >
> > + If users want to only focus on concept drift, it is straightforward to add another constraint to match the marginal distribution $\mathbb{P}(x)$ over the training and testing splits.
> >
> > ---
> > *"Similarly, it is difficult to reason about how much LS is guided by distributional shifts in the data or the predictive power of the predictor."*
> >
> > + In Figure 3, we compare the validation accuracy (a random 1/3 subset of the training split) and the test accuracy. The predictive power of the predictor is controlled for this comparison. In Figure 4 and 5, we demonstrate that there are distribution shifts over the learned training split and the testing split.
> >
> > ---
> > *"There are no details on how (3) is optimized. The free variables z_i are in {0,1}, which makes this not an direct application of gradient based methods as written. How is it done?"*
> >
> > + Our overall objective is end-to-end differentiable, as discussed in Section 3.
> >
> > + While the latent variables $z_i$ are in $\{0, 1\}$, the decision probabilities $\mathbb{P}_{Splitter}(z_i | x_i, y_i)$ are not. We defined the two regularizers $\Omega_1$ and $\Omega_2$ in Eq (1) over the decision probabilities, and therefore they are fully differentiable. In addition, in the first two paragraphs of Page 5, we included a step-by-step procedure on computing these two regularizers.
> >
> > + Please let us know if there are additional points of confusion.
> >
> > ---
> >
> > *"While i have not encountered work performing this exact task, a lot of work has focused on sources of bias. Also, there is another parallel line of work on test and evaluation of ML models that aims to uncover biases present in data collections."*
> >
> > + We have acknowledged 20+ recent work on de-biasing in our related work section. If there are other relevant pieces of work, please let us know and we would be happy to add the references.
> >
> > ---
> >
> > *"Algorithm 1 lacks any level of specificity in terms of how the sampler is used to split data and what optimization procedures are used to update the sampler or predictor. As such, I feel it is very difficult to reproduce this work."*
> >
> > + We respectfully disagree with the reviewer. Our submission includes both full implementation details and step-by-step instructions to run our code.
> >
> > + Algorithm 1 provides a high-level overview of the proposed method. In our submitted appendix, we have provided all details (4 pages in total) to reproduce our work. This includes details of the dataset, representation backbone, predictor and splitter architecture, optimization and training procedure.
> >
> > + Moreover, we've also provided a well-documented codebase for reproducibility. The repository includes scripts for generating the hard splits on all seven datasets in the paper. **Users can use our one-line API to generate challenging splits on any PyTorch dataset object.**

---

> > > ### Comment · Reviewer_Jd9v · 2022-11-16
> > > **Response to Rebuttal 2/2**
> > >
> > > 5.  Source of shift and Predictive Power of Predictor:  I have considered the response and looked again at the noted figures.  My comment is mostly focused on theoretic understanding, which I believe still holds.  These concepts are entangled in the algorithm and problem definition, and thus it is difficult to get a principled understanding of them.  However, I acknowledge there is some intuition and empirical findings that provide some insight into these questions.
> > >
> > > 6. z variables and differentiability: Thank you for the clarification.  This was an oversight on my part.  I recommend no changes with regards to this comment.
> > >
> > > 7. Reproducibility:  My comment is mostly focused on the main body of the submission, as the main submission itself is not sufficient for reproducibility.  I did not use this as a factor in my decision, and I acknowledge that the supplement is likely sufficient for reproducibility.  Thank you for pointing this out so others who find this paper know where to look.  I would still encourage the authors to provide some more details to Algorithm 1, as it is very high level to the point where not much can be gleaned from it.

---

> > > > ### Author Response · Authors · 2022-11-18
> > > > **Thank you for the speedy and detailed response! 4/**
> > > >
> > > > Thank you for the suggestion. We have updated Algorithm 1 to be more precise, and we also included a reference to the full pseudo-code in the Appendix.

---

> > > > > ### Comment · Reviewer_Jd9v · 2022-11-22
> > > > > **Response**
> > > > >
> > > > > Thank you so much for these detailed responses.  I am fairly confident that with these additional results, and a reframing the paper, I would argue for acceptance of this work.  However, I think it would require a considerable rewrite that is beyond what I would be comfortable with accepting for a camera-ready. I will increase my score to a 5 to reflect the strength of the rebuttal, and will discuss with the other reviewers how to come to a decision, given my hesitancy.

---

> > ### Comment · Reviewer_Jd9v · 2022-11-16
> > **Response to Rebuttal 1/2**
> >
> > Thank you for providing an extensive rebuttal.  I have a much better understanding of the work now.  There are a couple points I would like to address:
> >
> > 1. Practical Significance: I did not find this argument compelling.  Most related work that is cited in the paper focuses on training a model that generalizes to subpopulations in data.  The LS algorithm is designed to simply find if a subpopulation exists for which a model could be biased if training data is selected in a biased way.  In my opinion the former is of much more practical benefit.  That said, the Group DRO LS-split results are compelling, and in my opinion should be the focus of the paper.  Much of the paper emphasizes the importance of finding a biased split, but really the practical benefit is that LS can be used to train models with less bias without explicit group labels.  I would highly encourage a shift in focus for the paper.
> >
> > 2. Oracle baselines:  Thank you for clarifying.  I suppose what I am asking for is an ERM accuracy for the split chosen by LS and the split chosen by separating groups according to group labels, which, as I understand it, is not in Table 1.  Figures 4 and 5 show that the labels chosen may align with them, but doesn't give me a picture of how much the discrepancies between them affect performance under standard training conditions (ERM).  Further, I think it is important to be explicit as to what group labels were used to create the worst case.
> >
> > This actually makes me believe I am having trouble understanding Table 1, then.  As I understand it,  the results taken from Liu et al are from their Tables 1 and 7.  Table 1 uses the validation set group information to set hyperparameters, while Table 7 does not.  They do not include ERM in the latter setting, but this paper adds an additional row where training and validation group labels are unavailable.  It's not clear to me how the accuracies for the the ERM rows can be different unless either the mechanisms for chosing "worst" are different or ERM is using group labels in the validation set, which does not seem like standard ERM at that point.
> >
> > 3. Model without Learned Splitter:  The alternative I mentioned could move instances from the train set to the test set by simply also scoring training instances as well as test instances, similarly to how the proposed splitter considers all data.  I'm not convinced that all samples from the test set will necessarily score higher by a predictor than the train set, especially if you are early stopping (as stated in foot note 1).  I think there should either be some empirical rigor in determining that instances will never transfer between test and train sets for this alternative, or it should be considered a baseline, as it represents a simpler, more computationally efficient alternative.
> >
> > 4. Regularizers: So it seems that you are of the opinion that having an exact proportion as specified is not important for this problem, and you have convinced me as well.  I think you should explicitly state that, as it is most common for test sets to be predefined proportions of the full data set.

---

> > > ### Author Response · Authors · 2022-11-18
> > > **Thank you for the speedy and detailed response! 1/**
> > >
> > > *"Much of the paper emphasizes the importance of finding a biased split, but really the practical benefit is that LS can be used to train models with less bias without explicit group labels."*
> > >
> > > + Thank you for the helpful advice. We agree that ls’s role in training robust models is an important practical benefit. We will attempt to better position the “big picture” [below] in our exposition for later revisions and prioritize the practical benefit of ls in de-biasing.
> > >
> > > + We can break down debiasing into a two-step problem: 1) identify subpopulations, and 2) given these subpopulations, train a robust model.
> > >
> > >   + The majority of prior work has focused on the latter: *if* subpopulations have been annotated/identified by human experts, robust algorithms can be developed. In practice, however, these subpopulations are rarely readily available. While the number of tasks for which we would like to apply ML grows exponentially, only the most high-profile have such annotations (e.g. recent work ImageNetX).
> > >
> > >   + By focusing on the former, ls makes debiasing practical in real-world settings. Once ls identifies the subpopulations, training a robust model is straightforward.
> > >
> > > ---
> > >
> > > *"I am not convinced that intentionally splitting whole data sets to maximize a generalization gap necessarily means that training a model on the data set using standard data splits and procedures will be biased"*
> > >
> > > + We also want to clarify that even when the training data is not selected by ls, the standard ERM model can still be biased. Here we look at CelebA and use the official `train_data` and `valid_data`:
> > >   1. Learn Splitter to identify spurious splits **within** `train_data`.
> > >   2. Apply Splitter learned from step 1 to split `val_data` into [`val_data-train`, `val_data-test`].
> > >   3. Train Predictor using ERM on `train_data` (randomly held-out 1/3 of `train_data` for early stopping).
> > >   4. Evaluate Predictor on the original `val_data` and the two partitions `val_data-train`, `val_data-test`.
> > > + Results: training ERM on the standard splits failed to generalize on ls-identified subpopulations. We will include this result as an additional discussion in our paper.
> > >     | CelebA | `val_data` | `val_data-train` | `val_data-test` |
> > >     |:---:|:---:|:---:|:---:|
> > >     | average acc |  94.62% | 98.65% | 76.71% |

---

> > > ### Author Response · Authors · 2022-11-18
> > > **Thank you for the speedy and detailed response! 2/**
> > >
> > > *"Oracle baselines"*
> > >
> > > Thank you for explaining further. We are happy to provide additional results regarding this oracle baseline.
> > >
> > > + In CelebA, the label is `blond` vs. `not_blond` and the oracle bias attribute is `male` vs. `female`. Here we consider four different options to define the oracle splits.
> > >
> > >   | Split options   | train split                   | test split                    |
> > >   |----------|-------------------------------|-------------------------------|
> > >   | Oracle 1 | `male`                          | `female`                        |
> > >   | Oracle 2 | `female`                        | `male`                          |
> > >   | Oracle 3 | `male&blond` + `female&not_blond` | `male&not_blond` + `female&blond` |
> > >   | Oracle 4 | `male&not_blond` + `female&blond` | `male&blond` + `female&not_blond` |
> > >
> > > + Note that these oracle splits do not necessarily meet the regularity constraints that we proposed (see below).
> > >
> > >   | Split statistics | train split size | test split size | #`blond` in train | #`blond` in test |
> > >   |----------|:------------------:|:-----------------:|:------------------:|:-----------------:|
> > >   | Oracle 1 | 68261 (41.9%)    | 94509 (58.1%)   | 1387 (2.0%)       | 22880 (24.2%)    |
> > >   | Oracle 2 | 94509 (58.1%)    | 68261 (41.9%)   | 22880 (24.2%)     | 1387 (2.0%)      |
> > >   | Oracle 3 | 73016 (44.9%)    | 89754 (55.1%)   | 1387 (1.9%)       | 22880 (25.5%)    |
> > >   | Oracle 4 | 89754 (55.1%)    | 73016 (44.9%)   | 22880 (25.5%)     | 1387 (1.9%)      |
> > >
> > > + To measure the generalization gap for these oracle splits, we follow exactly the same strategy as Figure 1. That includes:
> > >   + Same predictor architecture and same hyper-parameter setting;
> > >   + Train the predictor using standard ERM.
> > >   + Hold-out 1/3 of the training split for early stopping
> > >   + Evaluate the average validation accuracy and the average testing accuracy.
> > >   + Report the average across 5 random seeds.
> > >
> > >   |          | val acc | test acc | test majority baseline |
> > >   |----------|--------------|---------------| --|
> > >   | Oracle 1 | 98.40        | 75.79         | 75.79% |
> > >   | Oracle 2 | 92.04        | 97.37         | 97.97% |
> > >   | Oracle 3 | 98.30        | 74.51         | 74.51% |
> > >   | Oracle 4 | 97.54        | 57.87         | 98.10% |
> > >
> > > + Observations:
> > >   + Oracle 1 & 3. The predictor suffered from the label imbalance during training. As a result, it behaves like the majority baseline at test time.
> > >   + Oracle 2. During training, the predictor learned to separate `blond` vs. `not_blond` for `female` (val acc 92.04). However, due to the domain shift (from `female` to `male`), the testing performance is slightly worse than the majority baseline.
> > >   + Oracle 4 has the biggest generalization gap. During training, the predictor learned to separate `male&not_blond` vs. `female&blond` very well using the gender bias (val acc 97.54). At test time, it failed to generalize.
> > >
> > > + The drastic difference between the statistics of ls-identified splits and these oracle splits makes it difficult to directly compare their generalization gaps. However, we agree with the reviewer that having this oracle comparison is beneficial. We will add this result as an additional discussion in our paper.
> > >
> > > ---
> > >
> > > *"It's not clear to me how the accuracies for the ERM rows can be different unless either the mechanisms for choosing "worst" are different or ERM is using group labels in the validation set, which does not seem like standard ERM at that point."*
> > >
> > > + The first ERM row is tuned using group labels from the validation set following [Liu et al., 2021]: "We tune the hyperparameters of all approaches based on worst-group performance on a small validation set with group annotations." This baseline is meaningful as it can help us understand whether the performance gain comes from a better validation criteria or the training method.
> > >
> > > + We have updated the caption of Table 1 to clarify this point.
> > >
> > > + The second ERM row section is tuned based on the average performance on the validation set. As we can see, there is a significant drop in terms of the worst-group performance (7.8% in Waterbirds, 6.1% in CelebA and 7.5% in MNLI).

---

> > > ### Author Response · Authors · 2022-11-18
> > > **Thank you for the speedy and detailed response! 3/**
> > >
> > > *"Model without Learned Splitter"*
> > >
> > > Thank you for clarifying. Here we consider two deterministic alternatives to the learned Splitter.
> > >
> > > + `ls w/o Splitter v1`: For each outer-loop iteration, we train the Predictor on the training split (with early stopping). Then we sort all input-label pairs (from both the training and testing split) based on $\mathbb{P}_{Predictor}(y_i | x_i)$, where $y_i$ is the ground truth label of input $x_i$. We put the lowest 25% in the testing split, leaving the top 75% in the training split.
> > > + `ls w/o Splitter v2`: Instead of sorting all input-label pairs across different label values (as in `v1`), we do it separately for different label values. For example in CelebA, we look at examples with label `blond` and put the lowest 25% $\mathbb{P}_{Predictor}(\texttt{blond} | x_i)$ in the testing split, and the top 75% in the training split. We repeat the same procedure for the other label value `not_blond`. This guarantees that the label distributions are matched between the training and testing splits.
> > >
> > > The `v1` variant easily collapses to a degenerative solution (label imbalance). The `v2` variant satisfies our regularity constraints by design (hardcoded).
> > >
> > > | CelebA split statistics     | `blond` in train | `not_blond` in train | `blond` in test | `not_blond`in test |
> > > |:--------------------:|:----------------:|:--------------------:|:---------------:|:------------------:|
> > > | `ls w/o Splitter v1` | 0                | 122078               | 24267           | 16425              |
> > > | `ls w/o Splitter v2` |  18200           | 103877               | 6067           | 34626               |
> > >
> > > Next, we look at the generalization gap and the running time produced by both `ls` and `ls w/o Splitter v2` on two datasets: CelebA and NoisyMNIST. In NoisyMNIST, the label of each input image has a 70% chance to be flipped to another random label.
> > >
> > > + `ls w/o learned Splitter` is faster, but the majority of the time cost still comes from Predictor's training.
> > > + `ls` consistently achieves the highest gap across the two datasets. While `ls w/o Splitter v2` is fine on CelebA, due to a fortuitous proportion of data minorities, `ls w/o Splitter v2` fails on NoisyMNIST, as this algorithm assumes that the percentage of minority groups is exactly 25% (which is hardly realistic). A learned Splitter can deviate from our prior based on the specific data circumstances, when necessary.
> > >
> > > |              | `ls` (CelebA)      | `ls w/o Splitter v2` (CelebA)  | `ls` (NoisyMNIST)      | `ls w/o Splitter v2` (NoisyMNIST) |
> > > |:------------:|:---------:|:----:|:----:|:----:|
> > > | Time on one A6000 (↓) | 2.3 hours | **1.8 hours** | 7.0 min | **4.5 min** |
> > > | train acc (↑) | 98.0 | **99.9** | **93.4** | 36.0|
> > > | test acc (↓)| **70.2** | 72.4 | 0.4 | **0.1**|
> > > | gap (↑) | **27.8** | 27.5 | **93.0** | 35.9 |
> > >
> > > Finally, we also note that `ls` allows users to detect whether a new incoming example belongs to the minority subpopulation (re: first CelebA experiment in part on of this response).
> > >
> > > Compared to `ls w/o Splitter v2`, `ls` also allows users to easily incorporate task-specific constraints. As we mentioned in the previous reply, users can combine their prior knowledge of the task to identify new biases (such as subpopulations that are independent of gender).
> > >
> > > ---
> > >
> > > *"Regularizers: So it seems that you are of the opinion that having an exact proportion as specified is not important for this problem, and you have convinced me as well. I think you should explicitly state that, as it is most common for test sets to be predefined proportions of the full data set."*
> > >
> > > Thank you for the suggestion. We have explicitly included this discussion as a footnote on Page 5.

---

### Official Review · Reviewer_Fe8j · 2022-10-23

**Confidence:** 4
**Correctness:** 3
**Technical Novelty And Significance:** 4
**Empirical Novelty And Significance:** 3
**Recommendation:** 8

**Clarity, Quality, Novelty And Reproducibility:**

I think this paper proposes a clever novel idea and method and is written clearly.

**Strength And Weaknesses:**

Strengths:
- Interesting and novel idea and approach, particularly the idea of using this bi-level optimization to identify biases in datasets.
- The paper is clearly written and provides a thorough empirical analysis of the proposed method.

Weaknesses:
- The algorithm can be computationally expensive, since it needs to retrain erm each time in the inner loop. Maybe it doesn't need to be trained to convergence? It would be interesting to see what would happen if each inner loop isn't trained fully to convergence. Another interesting experiment to see would be to alter the algorithm to just train the last layer in the inner loop (e.g. by pretraining the earlier layers with an initial ERM run), to reduce the computation requirements.

Not a weakness of this paper, but I would also be curious to see if the splitting could be used in other settings, such as OOD detection.

**Summary Of The Paper:**

This paper proposes using an adversarial bi-level optimization to find hard train/test splits of a dataset. More specifically, the inner loop corresponds to normal ERM training on that train/test split, and the outer loop learns a Splitter that changes the train/test split to make generalization in the inner loop difficult. The authors use two regularizers to avoid degeneracies: first, constraining the ratio of train/test dataset sizes, and second, constraining the label ratio in the two datasets (to avoid all labels being in the train split and all of the others being in the val split). This automatic splitting can then be used to identify biases in the dataset, like spurious correlations, and you can combine it with group DRO to get robust models.

**Summary Of The Review:**

The proposed idea is novel and interesting, so I recommend accepting the paper.

---

> ### Author Response · Authors · 2022-11-16
> **Thank you for your detailed comments!**
>
> *"The algorithm can be computationally expensive, since it needs to retrain erm each time in the inner loop. Maybe it doesn't need to be trained to convergence? It would be interesting to see what would happen if each inner loop isn't trained fully to convergence."*
>
> + Thank you for the suggestion. Indeed, we can gain a moderate speed boost with minimal loss in generalization gap.
>
> + For example, if we train the predictor for 100 steps vs. to full convergence on CelebA, we observe a slight drop in generalization gap (28.76% to 25.84%) with a significant decrease in runtime (30%). For larger applications, this could be computationally quite helpful.
>
>     | CelebA (resnet50) | Train for 100 steps | Train to full convergence |
>     |-----------:|:-----------------:|:----------------------:|
>     | Time (A6000)  | 1.6 hours       | 2.3 hours            |
>     | Validation acc | 94.14% | 95.34% |
>     | Testing acc | 68.30% | 66.58% |
>     | Generalization gap | 25.84% | 28.76% |
>
> + Minor note: the runtimes in this new table were produced by a more powerful server, so they are not directly comparable to Table 2.
>
> ---
>
> *"Another interesting experiment to see would be to alter the algorithm to just train the last layer in the inner loop (e.g. by pretraining the earlier layers with an initial ERM run), to reduce the computation requirements."*
>
> + We actually considered this approach during our algorithm development, but sadly it didn't work. :') On a high level, after pre-training with ERM, the final layer features were already well-separated for different label values. As a result, it became impossible for the Splitter to trick the Predictor.
>
> + Empirically, the generalization gap on CelebA was only 1.1% vs. 28.7% (proposed version of ls).

---

> > ### Comment · Reviewer_Fe8j · 2022-11-16
> > **Thanks for the response**
> >
> > Thanks to the authors for the response! I think the work is insightful and novel and will raise my score.

---

> > > ### Author Response · Authors · 2022-11-16
> > > **Thank you for the speedy update**
> > >
> > > Thank you so much. We're glad you enjoyed our paper!

---

### Official Review · Reviewer_GE6C · 2022-10-23

**Confidence:** 4
**Correctness:** 3
**Technical Novelty And Significance:** 3
**Empirical Novelty And Significance:** 3
**Recommendation:** 5

**Clarity, Quality, Novelty And Reproducibility:**

- clarity: good
- novelty as a task: great
- novelty as a technique:
     - the algorithm to partition the data is quite limited as it is a fairly straightfoward search through the dataset.
          - the two constraints at Eq 1 seem to be an over-complicated paraphrase of simple ideas (sufficient training samples and label balance)
          - line 7 and line 8 at algorithm 1 are random samples (that can probably be much improved with more heuristics, e.g., whether searched dataset will update the later search strategies)

- quality
     - it seems weird that eq.3 does not involve any hyperparameters to balance the weight between the main objectives and the regularization loss.
          - if this is a typo, then detailed discussion will be needed on the choice of the hyperparamters
          - otherwise, then probably more discussions will be needed on why such terms are not needed
    - the fact the group-DRO works on the new partition data seems to offer a piece of evidence that the method doesn't work that well.
         - group-DRO works by leveraging the minority of the samples during the training, however, if the algorithm successfully achieves its goal as depicted in Figure 1, there probably do not exist any minor samples in the training set for DRO to use
         - probably it's more convincing to test other algorithms that explicitly account for the bias, as can be useful without the usage of the minor samples.
    - it might be better to use the performances of these more advanced algorithms to evaluate the performance of the partition algorithm (and its variants, as ablation studies), then the detailed setup of all these algorithms will become essential.

- reproducibility:
     - given the limitation of the novelty of the method, it is probably necessary to discuss computing loads in details in the main manuscript (any evidence a more efficient algorithm is not even needed?)

- minor
     - the authors offer a good summary of papers explicitly discussing about biases, there are many other relevant papers following the debiasing strategy but do not use these keywords as this main ideas fall into a greater scope of ML robustness. It's probably better to expand the literature scope to a broader scope, e.g. (each of these represents a whole branch, instead of the listed individual paper)
          - Making the V in VQA Matter: Elevating the Role of Image Understanding in Visual Question Answering
          - Learning robust representations by projecting superficial statistics out
          - Invariant risk minimization

**Strength And Weaknesses:**

- strengths:
   - the paper studies a very new and interesting problem
   - figures/visualizations are stunning
   - the writings are very clear (although might be too detailed)

- weakness:
   - as a new study, many new questions are left unanswered for the community to follow in the later stage
   - while the visualizations are stunning, they look more like belong to brochures instead of academic papers, many figures do not necessarily need such visualization to explain. A simple table will do the job and then save a lot of space for more detailed discussions on other aspects.

**Summary Of The Paper:**

The paper studies a very interesting and novel problem setting: how to split the data in a way that when a model is trained on one partition cannot generalize to the second partition: while this task does not make realistic sense, careful studying of it will surely reveal many other properties that are important in the machine learning community. On the other hand, I feel like the actual algorithm proposed by this paper used for this new task is still in its fairly preliminary stage, with multiple questions unanswered.

**Summary Of The Review:**

Overall, I think this is a greatly interesting paper with a lot of potentials, but probably a bit preliminary at this moment.

---

> ### Author Response · Authors · 2022-11-16
> **Thank you for your detailed comments.**
>
> *"the algorithm to partition the data is quite limited as it is a fairly straightfoward search through the dataset."*
>
> + The number of possible splits is exponential with respect to the size of the dataset. A straightforward search cannot accomplish this goal.
>
> + In our algorithm, the splitter learns how to split the dataset so that the predictor cannot generalize. One can also think about the algorithm as a form of meta-learning, where the overall goal is learning to fool the predictor.
>
> ---
>
> *"the two constraints at Eq 1 seem to be an over-complicated paraphrase of simple ideas (sufficient training samples and label balance)"*
>
> + Indeed, these two constraints encompass very simple ideas, essential to preventing degenerate solutions. In our work, we chose to formulate these constraints as the KL divergence between the observed and prior distributions, which result in an objective that is end-to-end differentiable.
>
> + We would be happy to try any simpler suggestions that are likewise differentiable.
>
> ---
>
> *"line 7 and line 8 at algorithm 1 are random samples (that can probably be much improved with more heuristics"*
>
> + While leveraging heuristics makes sense at first glance, data-specific heuristics may not be available for all domains, and it can become an engineering/annotation rabbit hole to figure out the best heuristics for any given task.
>
> + Compared to previous de-biasing methods (e.g. group DRO and IRM), a benefit of this work is the lack of reliance on such heuristics, such as data environments or attribute labels.
>
> + Of course, if these heuristics are available, users can easily incorporate them into our method as additional regularizers.
>
> ---
>
> *"it seems weird that eq.3 does not involve any hyperparameters to balance the weight between the main objectives and the regularization loss."*
>
> + As we mentioned in our paper (last paragraph on Page 5), the unweighted summation works well out-of-the-box across all seven datasets.
>
> + Mathematically, it is worth noting that the three terms in Eq 3 are all log-losses over the splitter's decision probabilities. Compared to other regularization techniques (e.g. L1, L2 losses and gradient penalties), they operate on the same scale.
>
> + Moreover, the size of the minority group is naturally different for different datasets. We consider the two regularizers as our priors on the resulting splits, and we don't want to tune the weights to enforce these regularizers as hard constraints.
>
> ---
>
> *“the fact the group-DRO works on the new partition data seems to offer a piece of evidence that the method doesn't work that well”*
>
> + We would like to clarify a misunderstanding.
>
> + An added benefit of ls is that it trivially enables automatic debiasing: in lieu of human-identified groups, which must be manually conceived and annotated, one can apply group DRO out-of-the-box to ls-identified splits.
>
> + The fact that group-DRO works well with ls-identified splits (Table 1) offers positive evidence that ls separates minority groups from majority groups without additional human supervision. In contrast, applying group-DRO to random splits (which don't separate out minority groups) will not work.
>
> ---
>
> *“it might be better to use the performances of these more advanced algorithms to evaluate the performance of the partition algorithm (and its variants, as ablation studies), then the detailed setup of all these algorithms will become essential.”*
>
> + We used group-DRO on top of ls-identified splits for its simplicity and efficacy. In our paper, we demonstrated that when the bias annotations are provided for the validation data (second section), the group-DRO baseline already outperforms all recent state-of-the art de-biasing methods (CVar DRO, LfF, EIIL, JTT).
>
> + To reiterate from the previous question, the fact that simple group-DRO works exceedingly well is ample support for the difficulty of ls-identified splits.
>
> ---
>
> *“References for ML robustness"*
>
> + Thank you for the great suggestions. We will incorporate these areas into our related work.

---

> > ### Comment · Reviewer_GE6C · 2022-11-16
> > **response to rebuttal**
> >
> > Thanks for the clarification, I think there is a misunderstanding about my question regarding the effectiveness of group-DRO, as we both know that group-DRO works the best when there are still minor samples in the training set (in comparison to random split as the authors put in rebuttal). However, my point is, if the proposed algorithm works as expected, why are there still some minor samples in the training set, shouldn't all the samples of this property only appear in the test partition? (The other polarity of minor samples in contrast to the random split, for which the authors offer this rebuttal).

---

> > > ### Author Response · Authors · 2022-11-17
> > > **Thank you for the speedy response. Clarification of our de-biasing procedure.**
> > >
> > > Thank you for the speedy response and for clarifying the point of confusion!
> > >
> > > Each dataset has predefined splits `train_data`, `val_data`, `test_data`, the same as those used in previous literature. The results in Table 1 are reported on `test_data`.
> > >
> > > In Group DRO, [Sagawa et al., 2020] demonstrated that by using human-annotated attributes to define groups *within* `train_data`, they could significantly improve worst-group performance on `test_data`.
> > >
> > > ls only moves data points *within* splits, rather than *across*, i.e. `train_data` points will never swap with `test_data`. Any minority groups present in `train_data` will remain in `train_data`, and so on.
> > >
> > > Specifically, our automatic de-biasing procedure goes through the following steps (more details can be found on page 17 of the Appendix).
> > > 1. Learn Splitter on `train_data`, which is partitioned into `train_data-train` and `train_data-test`.
> > > 2. Use the learned Splitter to split `val_data` into `val_data-train` and `val_data-test`.
> > > 3. We apply group-DRO to minimize the worst-group accuracy within `train_data`. Instead of using human-annotated attributes to define groups, we define groups based on the ls-identified splits: {`train_data-train`, `train_data-test`}.
> > > 4. We measure the worst-group  accuracy over {`val_data-train`, `val_data-test`} for early-stopping.
> > > 5. We evaluate the standard worst-group accuracy (defined by human-annotated attributes) on the heldout `test_data`.
> > >
> > > We hope this makes more sense, and let us know if you have any remaining questions!

---

> > > > ### Comment · Reviewer_GE6C · 2022-12-08
> > > > **response to clarification**
> > > >
> > > > Thanks for further clarification, this answers my primary question, and I will update my rating.

---

### Official Review · Reviewer_pwSX · 2022-10-26

**Confidence:** 4
**Correctness:** 4
**Technical Novelty And Significance:** 4
**Empirical Novelty And Significance:** 4
**Recommendation:** 6

**Clarity, Quality, Novelty And Reproducibility:**

The manuscript is logically well-written and easy to follow. Also, to the best of my knowledge, this is the first paper to successfully deal with detecting biases in datasets without bias annotations.

**Strength And Weaknesses:**

Strength

1. The paper is logically well-written, including a discussion of the limitations of ls. In particular, additional experiments related to label noise further enhanced the reliability of the paper.

2. Several experiments have well validated the effect and applicability of ls. In particular, the experimental results using GropuDRO with ls are impressive. It would be interesting to conduct experiments with ls applied to other methods using bias annotations in the validation phase.

3. This research tackles an important question, and I believe it is a helpful research direction for the community.

Weakness

1. I think there should be further experiments and discussion on how the \delta value affects the results of ls. Therefore, during the review period, I hope the authors validate that the model works robustly even with various \delta values. Also, it would be helpful to understand ls if the authors address how the model behaves when the percentage of minority groups varies rather than 25%.

2. As the authors mentioned in Section 5, ls seems to have issues with scalability. To further discuss this concern, I think the training time of other debiasing methods should be provided in Table 2 of the Appendix.

3. In Table 1, the source of the results brought from the previous paper is omitted. That information should be included to determine whether the experimental settings of ls are the same as those of the baselines.




**Summary Of The Paper:**

This paper proposes Learning to Split (ls), a novel algorithm that automatically detects a potential bias in datasets. More specifically, ls consists of a Splitter and a Predictor. The Splitter learns to divide the dataset into a train split and a test split. Then, the Predictor is trained using the divided train split and measures the generalization error with the test split. The Splitter is then trained in the direction in which the generalization error increases. That is, ls find a train split that cannot generalize well in the test split and then observe the bias existing in the dataset. The authors validated that ls behaves as expected on datasets with bias annotation. Finally, when ls is combined with debiasing methods that require bias annotation during training, such as GroupDRO, these models were trained successfully without bias annotation

**Summary Of The Review:**

The main contribution of this paper is that potential biases present in datasets can be detected without bias annotations. Although there are concerns regarding scalability, I believe it is valuable as the first paper that automatically detects potential biases.

---

> ### Author Response · Authors · 2022-11-16
> **Thank you for your detailed comments.**
>
> *"I think there should be further experiments and discussion on how the \delta value affects the results of ls. Therefore, during the review period, I hope the authors validate that the model works robustly even with various \delta values."*
>
> In the ablation studies (pages 17-18), we showed that if we completely remove $\delta$ (regularizer $\Omega_1$), ls is still able to identify challenging splits (59% gap in Beer Look and 54% in Tox21). Though by design, the size ratios are less stable (89%/11% in Beer Look and 69%/31% in Tox21).
>
> ---
>
> *"Also, it would be helpful to understand ls if the authors address how the model behaves when the percentage of minority groups varies rather than 25%."*
>
> Our regularity constraints are formulated as soft regularizers, i.e. there is no requirement that minority groups constitute exactly 25% (which itself was an arbitrary threshold based on literature). For example when using ls for label noise detection (pages 18-20), our default $\delta$ value (25%) works well even when we vary the percentage of noisy labels from 10% to 80%.
>
> One can think of these regularizers are priors on the resulting splits. The actual posteriors can be very different, depending on the datasets.
>
> ---
>
> *"As the authors mentioned in Section 5, ls seems to have issues with scalability. To further discuss this concern, I think the training time of other debiasing methods should be provided in Table 2 of the Appendix."*
>
> Here is the runtime of all other de-biasing methods on CelebA. Note that for even the same method, the resulting times can be very different for different hyper-parameter configurations.
>
> |            | ERM | CVaR DRO | EIIL | JTT |
> | ---------- |---------------|------|------|---------------|
> | single run (V100) | 0.1-0.3 hour |  0.5-1.1 hours | 0.8-1.6 hours | 0.8-1.5 hours |
>
> While it took more total time (5.6 hours) for ls to identify the spurious splits, the ls procedure only needs to be run once. Factoring in the time required for other baselines to perform hyper-parameter search, we consider the overhead of running ls acceptable in our experiments.
>
> In practice, as reviewer Fe8j has pointed out, the inner-loop doesn't have to be trained till full convergence. In CelebA, this can significantly reduce the time cost by 30% while still being able to generate challenging splits.
>
> ---
>
> *"In Table 1, the source of the results brought from the previous paper is omitted. That information should be included to determine whether the experimental settings of ls are the same as those of the baselines."*
>
> Thank you for the suggestion. We have included the source of the results in Table 1.

---

> > ### Comment · Reviewer_pwSX · 2022-11-25
> > **Response**
> >
> > Thanks to the authors for their responses. Specifically, the responses addressed my concerns regarding the delta values well. However, my concern about scalability still remains. Therefore, I would like to keep my score.

---

### Author Response · Authors · 2022-11-16
**Summary of our contributions**

Thank you all for your detailed comments and suggestions!

Here is a summary of the major contributions of this work.

1. We propose learning to split (ls), an algorithm that **learns to partition the dataset such that downstream predictors cannot generalize across splits.**
2. Empirical results on text classification, natural language entailment, image classification and molecular property prediction show that ls generates astonishingly challenging splits that correlate with human-identified biases.
3. ls enables automatic debiasing trivially: applying group-DRO to ls-identified splits outperforms previous SOTA by 23.4% (worst group accuracy) across three benchmarks.
4. ls is task-agnostic and can be applied to any supervised learning tasks. Our algorithm is released as a plug-and-play package. Users can easily use our one-line API to split any PyTorch dataset object.

---

### Decision · Program_Chairs · 2023-01-20

**Decision:**

Reject

**Justification For Why Not Higher Score:**

Presentation did not put the emphasis or expansion on the actual valuable contribution, and some parts were over-complicated-- significant work would be needed to make this suitable for publication.

**Justification For Why Not Lower Score:**

N/A

**Metareview: Summary, Strengths And Weaknesses:**

The paper proposes a procedure to split a dataset to make generalization from one to the other difficult.
The reviewers appreciated that the proposal was simple and that some of the ideas were novel and is an interesting first work to identify biases in the dataset.
However, during the virtual discussions, some of the reviewers agreed that while the paper has some meaningful contributions, the paper would need significant work to (1) better highlight the relevant contributions and its impact, which was only a small part in the paper submission, and to (2) simplify the presentation of the paper as it seemed to some of the reviewers that some key parts were intentionally over-complicated.

**Summary Of Ac-Reviewer Meeting:**

Jd9v and GE6C attended. pwSX couldn't attend due to time zone differences. The rest were unresponsive.

Jd9v: questions the basic premise of taking full dataset and taking a biased split is a useful task. There was a back-and-forth.
For purpose of training of later model was useful, however  entire paper would need to be rewritten since this was only a small part of the paper.
Ultimately, significant rewrite would be needed for paper to be ready to be submitted. Difficult to find impactful contribution in the paper as presented.

GE6C: Agree. Work itself is interesting but writing seems to raise some concerns. Especially authors agree that authors intentionally made presentation more complicated than necessary.